# Application of Plant Proteases in Meat Tenderization: Recent Trends and Future Prospects

**DOI:** 10.3390/foods12061336

**Published:** 2023-03-21

**Authors:** Syahira Izyana Mohd Azmi, Pavan Kumar, Neelesh Sharma, Awis Qurni Sazili, Sung-Jin Lee, Mohammad Rashedi Ismail-Fitry

**Affiliations:** 1Department of Food Technology, Faculty of Food Science and Technology, Universiti Putra Malaysia (UPM), Serdang 43400, Selangor, Malaysia; izyanaazmi@gmail.com; 2Department of Livestock Products Technology, College of Veterinary Science, Guru Angad Dev Veterinary and Animal Sciences University, Ludhiana 141004, Punjab, India; pavankumar@gadvasu.in; 3Institute of Tropical Agriculture and Food Security, Universiti Putra Malaysia (UPM), Serdang 43400, Selangor, Malaysia; 4Division of Veterinary Medicine, Faculty of Veterinary Sciences and Animal Husbandry, Sher-e-Kashmir University of Agricultural Sciences and Technology of Jammu, Ranbir Singh Pura 181012, Union Territory of Jammu and Kashmir, India; drneelesh_sharma@yahoo.co.in; 5Department of Animal Science, Faculty of Agriculture, Universiti Putra Malaysia (UPM), Serdang 43400, Selangor, Malaysia; awis@upm.edu.my; 6Halal Products Research Institute, Universiti Putra Malaysia (UPM), Serdang 43400, Selangor, Malaysia; 7Department of Applied Animal Science, College of Animal Life Sciences, Kangwon National University, Chuncheon 24341, Republic of Korea

**Keywords:** plant protease, novel protease sources, ultrasound, high-pressure processing, immobilization, meat quality

## Abstract

Papain, bromelain, and ficin are commonly used plant proteases used for meat tenderization. Other plant proteases explored for meat tenderization are actinidin, zingibain, and cucumin. The application of plant crude extracts or powders containing higher levels of compounds exerting tenderizing effects is also gaining popularity due to lower cost, improved sensory attributes of meat, and the presence of bioactive compounds exerting additional benefits in addition to tenderization, such as antioxidants and antimicrobial effects. The uncontrolled plant protease action could cause excessive tenderization (mushy texture) and poor quality due to an indiscriminate breakdown of proteins. The higher cost of separation and the purification of enzymes, unstable structure, and poor stability of these enzymes due to autolysis are some major challenges faced by the food industry. The meat industry is targeting the recycling of enzymes and improving their stability and shelf-life by immobilization, encapsulation, protein engineering, medium engineering, and stabilization during tenderization. The present review critically analyzed recent trends and the prospects of the application of plant proteases in meat tenderization.

## 1. Introduction

Meat tenderness plays a decisive role in determining the eating qualities of meat (juiciness, mouthfeel, palatability, flavor, texture, and appearance) and is regarded as one of the most important properties of cooked meat [1,2]. Consumers would most likely consider purchasing meat that is guaranteed to be tender [3]. Tenderness is one of the factors that influence the overall palatability and acceptance of meat apart from flavor and juiciness [4]. Furthermore, it proves a motivational factor for repeated purchases even at a higher price, and poor meat tenderness results in significant losses to the meat industry [5]. The components involved in meat tenderness are the connective tissue, sarcomere length, post-mortem changes, proteolytic potential, and degradation of the myofibrillar proteins [6,7]. Tenderness is the primary concern for red meat (beef, lamb, pork, and lamb) due to its production practices and age at slaughter. 

Meat with high toughness/lower tenderness becomes hard and has a higher cooking loss and is less juicy. It has poor palatability and chewability [8]. Tender meat is usually characterized by the appearance of marbling (intramuscular fat) whereas tough meat has a higher content of connective tissues and myoglobin, thereby imparting intense red color to such meat. On shear force value measured by Warner–Bratzler shear force value (WBS), beef with a WBS 30 N or less is regarded as tender, and a value more than 46 N WBS confirms beef as tough, whereas, for wild deer, the above values for tenderness and toughness are 16 N and 22 N, respectively [9]. It has affected the overall meat production pattern, as Marquer et al. [10] observed a 7% decrease in meat production from bulls and a 4% increase in the meat produced from veal in the European market during 2009–2014. This increasing slaughter of young animals without harvesting their full meat production potential decreases yield and sustainability.

The consumer acceptance and marketability of low-value and tough meat cuts can be improved by using suitable tenderization mechanisms. The meat tenderization process and application of exogenous proteases have been covered in previous reviews [5,7,11,12,13,14,15,16,17,18]. The present review critically analyzed various aspects of the application of plant proteases such as novel sources, recent technologies, and future prospects in meat tenderization.

The wide variations and inconsistency in the tenderness of the same meat cuts from different batches, especially in premium and valuable meat cuts, remain a major concern for the meat industry and consumers [7]. Post-mortem meat tenderization under controlled conditions by using exogenous plant proteases could be a potential solution to this variability in meat tenderness, improving the tenderness of low-value meat cuts, and reducing the meat industry’s waste [19]. The present review provides an overview of the application of exogenous plant proteases in meat tenderization.

## 2. Meat Tenderization

Aging or conditioning refers to holding carcasses above their freezing point in the absence of microbial spoilage. During aging, various processes occur leading to the tenderization of meat such as the degradation of collagen [20], changes in sarcomere length during rigor mortis [21], the reduction of the diameter of muscle fiber bundles [22], and various chemical and structural changes in myofibrillar proteins [5]. Under aging/conditioning, meat tenderization takes place through the action of endogenous proteolytic enzymes such as calpains and cathepsins. The amount of these enzymes varies with the age and breed of the animals [23]. Calpains are cysteine endopeptidases present in cytosol around the Z-line. These can degrade myofibrillar proteins except for myosin and actin. Cathepsins B, H, and L are cysteine endopeptidases present in lysosomes and can degrade myofibrillar proteins. Other peptidases that could contribute to tenderization are 20S proteasome, caspases, di- and tripeptidyl peptidases, carboxypeptidase A, B, and aminopeptidases [24]. Aging or conditioning improves tenderness but not to the extent of effectively reducing the toughness of meat from old animals. This warrants the use of an appropriate tenderization process, such as using various technological interventions to degrade the connective tissues and myofibrillar protein.

The timeline for meat tenderness development is presented in Figure 1.

The most common methods of meat tenderization are physical/mechanical, chemical, and biological processes [26]. Mechanical processes such as the suspension method, electrical stimulation, tumbling, grinding, pounding, and blade tenderization are used in the industry [27]. Chemical methods of meat tenderization such as marinades are rapid and less labor intensive. Early Mexicans used to package meat by wrapping it in papaya leaves for tenderization. Tenderization of the meat is also carried out using marinades with needles that inject it into the carcasses or its deeper parts or by using a meat tumbler. Lastly, biological methods use intrinsic enzymes during the aging process or exogenous enzymes are added to the meat for meat tenderization [28]. The exogenous enzymes used in treating meat samples are usually injected, infused, or marinated into the meat to hasten the tenderization process [2].

These chemical methods of tenderization are not very effective for very tough meat. Tenderizers can also make the exterior of the meat mushy while leaving the interior tough and may impart undesirable flavor. The meat industry is undertaking several technological innovations to improve the efficiency of these methods by improving the delivery and efficiency of enzymatic action present in the marinades such as hydrodynamic-pressure processing, high-pressure processing (HPP), pulsed electric field, ultrasound, SmartStretch™, and Pi-Vac Elasto-Pack^®^ system [29].

## 3. Plant Protease as a Natural Meat Tenderizer

Based on the source of production, exogenous proteolytic enzymes can be categorized as plant proteases/proteinases, fungal proteases, bacterial proteases, and animal proteases. Proteases are present in all plants. These are vital for the plant’s physiological and development process [30]. Recently, the application of exogenous enzymes for meat tenderization has become an increasing focus of interest among food technologists and meat scientists [12]. The extraction, identification, and purification of plant proteinase from novel non-conventional sources have the potential to significantly contribute to the future supply of commercial exogenous plant proteinase.

Out of all the commercial enzymes used in the food industry, 60% were reported from proteases, while the remaining 40% were from other enzymes [31]. The meat tenderizer added for meat tenderization usually comprises a mixture of plant enzymes and bacterial collagenase [32]. Plant proteases are naturally produced and present in most plant sources compared to microbial proteases, which are mostly by-products of microbial fermentation [2]. In addition, the hydrolytic activity in myofibrillar protein by plant proteases is higher than the bacterial proteases [6].

Consumers also prefer plant proteases to be used instead of animal proteases due to the potential risk of transmission of illness, environmental concerns, sustainability, lower price, and the absence of animal welfare, religious, and ethical issues [15]. In some countries, the enzymes prepared by recombinant technology are not permitted in the food industry [15]. Furthermore, an additional advantage of these proteases from plant sources is the lack of any religious issues (such as halal compliance issues) compared to proteases from animals or microbial fermentation [33].

Plant proteases are already proven to improve meat tenderness by the mechanism of proteolytic degradation. The common exogenous plant proteases used by meat tenderization are bromelain, papain, ficin, actinidin, and zingibain. These proteases break down the muscle proteins in the connective tissue by hydrolyzing the peptidic linkages in the proteins into peptides and finally into amino acid components, thereby reducing the meat’s toughness [12,32].

Enzyme kinetics (V_max_, K_m_, and K_cat_) indicate the overall enzymatic reaction pattern. V_max_ refers to the maximum rate of reaction achieved when the enzyme is saturated with 0the substrate. Furthermore, the reaction rate and substrate concentration are affected by the enzymes’ affinity with the substrate, which is commonly represented as the Michaelis constant factor (K_m_) of the enzymes. K_m_ represents the concentration of substrate that permits the enzyme to achieve half V_max_. A higher K_m_ indicates a lower affinity and vice-versa [34]. K_cat_, derived from V_max_, refers to the maximum number of substrate molecules that are converted to product per active site per unit of time in a condition when an enzyme is saturated with the substrate [35,36].

Plant protease has good enzymatic efficiency as indicated by their kinetics and thermodynamics (Table 1). Chymotrypsin has a K_cat_ value of 100 per second [36]. Bromelain sourced from *Ananas comosus* was observed to have 0.777% K_m_ and 3.969 U/min V_max_ as calculated using the Lineweaver Burk curve by Nurhidayat et al. [37]. Papain enzyme extracted and purified from *Carica papaya* leaves was observed to have a K_m_ value of 1.47–8.70 mg/ mL and a V_max_ value of 0.42–0.4167 µmol/mL/min [38]. K_m_ of ficin for hydrogen peroxide substrate was reported at 0.35 mMol where V_max_ was recorded at 4.69 µg/mL [39].

A brief description of the physiological attributes of the plant proteases used for meat tenderization is depicted in Table 1.

Cysteine proteases, also known as thiol proteases, degrade proteins. This enzymatic action by cysteine proteases is caused by hydrolysis of the peptide bonds involved in the activation of a cysteine residue by a histidine residue, and both are present in the active site of the enzyme. It is commonly present in fruits such as papaya, fig, pineapple, and kiwi fruit [17].

Figure 2 shows the reaction mechanism of hydrolysis of peptidic linkage by cysteine protease. The process starts with protein abstraction of the cysteine and histidine residue to form nucleophiles. The nucleophile will react with the substrate and form a covalent tetrahedral intermediate. The histidine in the nucleophile will donate a proton to the peptide leaving the group in the substrate, causing the peptide leaving the group to cleave and hydrolysis of the peptidic linkage to occur. The cysteine will then undergo reprotonation and regenerate for further use. Breaking down this peptidic linkage will further degrade muscle fibers and improve tenderness [44].

Plant extracts are increasingly used as a natural preservative in the meat industry as a green alternative to synthetic preservatives [45,46]. The plant extracts are rich sources of polyphenols, essential oils, minerals, and other bioactive compounds [47,48,49,50,51,52]. Thus, utilization of these plant extracts could provide more benefits as compared to commercial enzymatic preparation, such as higher nutritive quality, enhanced oxidative stability and microbial quality, sustainability, economical price, variety of products, and a positive effect on the organoleptic attributes of meat products.

### 3.1. Papain

Papain (molecular weight-23,000 Da, 212 amino acids) is a cysteine protease extracted from papaya latex (EC 3.4.22.2). Crude papain can be retrieved by scoring the papaya fruit, allowing the latex to flow out and dry on the fruit [6]. Apart from the papain enzyme, other endopeptidases such as chymopapain (EC 3.4.22.6), caricain (EC 3.4.22.30), and glycyl endopeptidase (EC 3.4.22.25), can also be found in the latex [12].

Papain is one of the most commonly used exogenous plant enzymes used for meat tenderization due to its ability to break down both myofibrillar proteins and connective tissues. Based on analysis of the loss of protein denaturation peak during differential scanning calorimetry, Shang et al. [53] concluded that the increasing tenderness of puffer fish muscles correlate with the degradation and fading of the myosin heavy chain and actin bands and the formation of low molecular weight peptides (25–35 kDa). When papain concentration was increased, there was a relative decrease in the content of *α*-helix and an increase in irregular coil structure and disordered structure, conferring hydrolysis of mainly myosin and actin into low molecular weight peptides.

Papain application significantly reduced the shear force of beef stored at 5 °C for 2 weeks by releasing hydroxyproline due to the degradation of collagen and hydrolysis of myofibrillar proteins [54]. Papain can survive in a wide range of temperatures and pH (3–9), making it the perfect choice to be used in different industries. It has very high tenderizing potential and a little amount of it is sufficient for the tenderization effect (less than 0.01 AU/100 g), otherwise it could lead to a mushy structure due to its broad specificity and indiscriminate breakdown of proteins [2,55]. Papain can remain active and contribute to changes in product texture even after cooking [56]. To control the excessive reaction of papain, it is added into marinades and processed under controlled conditions. Alternatively, ascorbic acid may be added to control papain action in meat, but a higher level of ascorbic acid incorporation may further deteriorate flavors. Papain is not used to prepare premium quality meat products due to reduced juiciness score and perceived bitterness. This bitterness is developed due to the formation of bitter peptides during enzymatic proteolysis [57].

The purification process is crucial in extracting papain as it removes undesirable compounds that may interfere with the papain’s activity. Papain is traditionally extracted from papaya using precipitation methods; however, this method is not able to achieve a desirable purification percentage (below 40%) [58]. Papain thermostability as compared to the free enzyme could be further enhanced by immobilization on choline-lactic acid (ChCl-Lac) deep eutectic solvent treated with chitosan [59]. Aqueous two-phase system (ATPS) extraction is used to clarify, concentrate, and purify the enzyme extracted simultaneously in a single process. This method can be applied to papain and other plant proteases extraction [60]. Furthermore, by applying three-phase partitioning (TPP) extraction methods, papain with improved enzymatic activity (134%) with an optimum activity range from activity at 50 °C and pH 6.0 were observed [41].

Papaya extract (0.3% *w*/*w*, 5 h) in combination with HPP (350–450 MPa, 10 min, 4 °C) significantly improved the tenderness and food safety of chicken breast [61]. The authors reported a 6 log reduction in *Salmonella* spps. and severe degradation of muscle fibers and connective tissues in chicken breast meat. Soaking buffalo meat in pineapple fruit extract improved the tenderness and WHC and reduced the cooking loss or shrinkage after cooking [62].

### 3.2. Bromelain

Bromelain is made up of 285 amino acids and a high content of alanine and glycine. It is a mixture of cellulase, peroxidase, phosphates, protease inhibitors, and glycoproteins. It is extracted from pineapple (*Ananas comosus*) mainly from the stem (EC 3.4.22.32), and fruits with less concentration are also present in pineapple wastes (EC 3.4.22.33 for leaves, cores, and peels) [63]. Commercially, bromelain is used in meat and other industries such as baking, and stabilizer (in beer production). It is used in the pharmaceutical industry as an anti-tumorigenic agent, anti-inflammatory, and anti-thrombotic, and can prevent diseases including bronchitis, sinusitis, and thrombophlebitis. Bromelain improves the extraction yield and gel strength from bovine skin with extracted samples having the presence of large particle sizes, and irregular and dense networks [64]. Bromelain is used for meat tenderization, and freeze-dried bromelain powder was reported to improve the tenderness and nutritive value of tough beef by decreasing shear force and increasing amino acid content in the treated meat [65].

Bromelain enzymatic activity and performance are affected by pH and temperature. Truc et al. [66] studied the effect of pH and temperature on bromelain activity and optimality. As a result, the optimality of the bromelain enzyme varies with temperature and pH. In acidic conditions, bromelain enzyme activity is optimum at the temperature range of 10–20 °C, while in alkaline conditions, the temperature for optimum bromelain activity is 30–40 °C. Besides that, at neutral pH values, a temperature of 40–60 °C is required to achieve the optimum activity [67].

Bromelain has a tenderizing effect by degrading the myosin light chain and troponin T without affecting actinin, resulting in the generation of protein fragments with smaller sizes [68]. However, its broad specificity also causes a mushy texture if the enzymatic action is not properly controlled. Pear fruit cysteine protease has been proposed as a potential alternative to bromelain to overcome the problem of a mushy texture [69]. The pH of meat and other processing parameters significantly affected the bromelain’s tenderizing efficiency. Kamarul Zaman observed the decreased hardness, water holding capacity (WHC), moisture content, and redness (*a** value) of Brahman beef upon bromelain tenderization [65]. Bromelain application also increases the nutritive value of meat by increasing its essential amino acids content [70], which could be due to preferential cleavage sites of bromelain at lysine, glutamic acid, glycine, methionine sulfoxide, and alanine [71].

Various technologies, such as chromatography, microfiltration, ultrafiltration, centrifugation, lyophilization/freeze-drying, reverse micellar extraction (RME), and cross-linkings are applied during extraction and purification of bromelain to improve purification efficiency and activity. Bromelain extracted from pineapple pulp using the ultrafiltration–centrifugation–lyophilization process had optimum enzymatic activity at 55–59 °C and pH 6.5–7.5 [72]. Bromelain extracted using the aqueous two-phase system (ATPS) formed using ATPS formed by polyethylene glycol (PEG) and polyacrylic acid had a high purification factor (25.78) and a purification fold of 4.0 [73]. Bromelain extracted from a pineapple core using reverse micellar extraction (RME) was noted to improve bromelain activity (85%) and purification fold (5.2), resulting in a higher degree of protein fragmentation and tenderization [74]. Banerjee et al. [75] incorporated a cross-linking process in the extraction of bromelain (60% *w*/*v* ammonium sulfate) by using glutaraldehyde (80 mM glutaraldehyde for 4 h) for stabilizing it as insoluble cross-linked aggregates. The extracted cross-linked bromelain had high heat and pH stability and high activity retention (>85% after five cycles of usage) [75].

### 3.3. Ficin

Ficin is a popular plant protease used in the tenderization of meat, mainly by hydrolyzing peptide bonds at aromatic residues. Ficin contains several proteases (glycoproteins, four types of ficin as A, B, C, and D) extracted from the latex of the fig (*Ficus carica*). Ficin has the properties of an anti-parasite and hemostatic agent [6]. Apart from the meat industry, ficin can also be used in the dairy industry due to its ability to facilitate milk coagulation [76]. Ficin is reported to optimally act at pH 8.0 and 60 °C temperature. Besides, it is stable at pH ranging from 2.0–8.0 and at 4 °C temperature for 20 h [15]. Earlier research found that ficin is thermally stable even until the temperature of 70 °C.

Current research on ficin extraction is limited. Researchers tend to use commercially available ficin from the market for research purposes, for example, Li et al. [77] used ficin manufactured by Tokyo Chemical Industry Co., Ltd. From Tokyo, Japan, Zhang et al. [78] used ficin from Henan Shengside Ltd., Henan, China, while Sullivan et al. [79] used ficin manufactured by Enzyme Development Corp., New York, NY, USA. From commercial preparations, ficin extraction is created using salt fractionation and chromatography using carboxymethyl cellulose. The precipitation of ficin with ammonium sulfate after gel filtration in a gel filtration medium, Sephadex G-100, has been undertaken to precipitate ficin [80]. The purified ficin was observed to be susceptible to autolysis when stored at high temperatures, and the rate of degradation increases when the storage time increases.

Various studies investigated the effectiveness of ficin on various types of meat, viz., camel meat, beef, and pork [78,81]. Sullivan et al. [79] reported that treating beef, particularly the *triceps brachii* and *supraspinatus* muscles, with ficin resulted in improved tenderness due to the myofibrillar and collagen degradation. On the other hand, camel round muscle was treated with ficin at 100 ppm with a storage temperature of 4 °C for 4 days and after that, results showed a decrement in terms of hardness, springiness, cohesiveness, gumminess, chewiness, and shear force, thus improving the tenderness of camel round muscle.

Thus, plant proteases can be used to improve the tenderness of various types of meat such as beef, mutton, camel, rabbit, squid, pork, buffalo, chicken, and yak. This shows that plant proteases are versatile and can be used in meat industries. Other than that, it can also be seen that the treatment with proteases resulted in a positive impact on the tenderness; however, some of the research also resulted in drawbacks, which are reduction of WHC, juiciness, and degradation of color. Some researchers were able to come up with a solution for these drawbacks: combining the enzyme treatment with HPP. Some plant proteases require secondary treatment to tenderize the meat ideally without degrading other quality.

## 4. Miscellaneous Plant Proteases

The food industry is continuously searching for newer, sustainable, economical plant proteases with improved efficiency and immobility. Most of these are not commercially approved and do not have a generally recognized as safe (GRAS) status. Still, a lot of research is undertaken to assess their utilization in the food industry and food safety. Actinidin extracted from kiwifruit, zingibain, and curcumin are important plant proteases that have been increasingly explored by the meat industry for obtaining desirable tenderization. Some plant proteases that have considerable potential in meat tenderization are as follows.

### 4.1. Actinidin

Actinidin is a major protease in kiwifruit/Chinese gooseberry (*Actinidia deliciosa*) [82]. Kiwifruit actinidin (EC 3.4.22.14) can break down protein and improve digestion in the small intestine, specifically in the ileum part. Kiwifruit has attracted interest and usage in the food industry due to its food protein-digesting ability [83]. A pharmacopeia from Tang Dynasty listed that the *Actinidia* species, which includes kiwifruit, could improve digestion, reduce irritation, and cure dyspepsia and vomiting [82]. Xuxiang cultivar had a high amount of actinidin with the highest activity among common cultivars, viz., Jinxiang, Hayward, Jinlong I, and Qihong grown in China [2]. Its protease consists of 50–60% of soluble proteins. The optimum pH and temperature for purified actinidin activity derived from the Xuxiang cultivar are stated to be 3.5 and 40 °C, respectively [2]. The major concern regarding the application of actinidin is its allergenicity in some sensitive children and adults, leading to itchiness or a tingling feeling around the mouth and anaphylactic reactions [84,85]. This has limited the commercial application of actinidin in the food industry.

Actinidin is extracted and purified from kiwifruit using a *N*-α-CBZ-lysine *p*-nitrophenyl ester or potassium phosphate buffer and a centrifugation process. Kakash et al. [86] extracted actinidin from peeled kiwi flesh using a potassium phosphate buffer and a centrifugation process at 4000 rpm at 4 °C for 10 min [86]. The whole extraction process was conducted in cold conditions/ice to minimize the enzymatic activity during extraction [87]. The extracted actinidin resulted in enzyme activity of 0.9 U/mg. Zhu et al. [88] prepared kiwifruit extract from green kiwifruit before treating meat brisket samples. The kiwifruit was peeled, purified, and filtered using muslin clothes to obtain pulp, followed by the addition of ice-cold sodium phosphate buffer and centrifugation.

Several studies have reported the tenderizing effect in meat by applying actinidin. Actinidin at 0.5 mg/100 g of muscles was reported to decrease the shear force of pork and rabbit muscle [2]. Zhu et al. [83] reported that treating brisket steaks with 5% of a 3 mg/mL solution of commercial actinidin extract (Actazin™ from Anagenix Ltd. Auckland, New Zealand) and subsequently undergoing vacuum tumbling and a sous vide cooking process for 30 min at 70 °C resulted in the most acceptable and favorable outcomes. No noticeable changes were reported for the pH, color, and cooking loss between the untreated and treated samples; however, the treated sample had higher sensory scores regarding tenderness, juiciness, and flavor. It was also reported that the best method to incorporate this enzyme into meat treatment is by using the injection method because the tenderizing effect will be more uniform compared to the marination method.

Actinidin tends to over-tenderize the meat if it is not inactivated as it will keep initiating hydrolysis of the protein in the meat and eventually result in a mushy and unfavorable texture of the meat [83]. Therefore, Zhu et al. [83] studied tenderization of a beef brisket using a commercial kiwifruit enzyme and reported that the amount of enzyme and enzyme inactivation technique used affected the texture of the meat. The lower the concentration of enzyme, the lower the tendency of the meat to become mushy. Moreover, injecting 0.2 mL of the enzyme into the injecting sites using fine needles would prevent the mushy texture, but still tenderizing the meat and continuing the process with vacuum tumbling could better distribute the injected enzyme [83]. Thus, actinidin does not affect the post-treatment beef brisket’s pH, color, and juiciness and slightly improves the flavor.

Treating beef with actinidin improves the sensory scores for tenderness, juiciness, flavor, overall liking, and rating [89]. However, Biffin et al. [90] observed that treating alpaca meat with actinidin reduces consumer acceptance when compared to untreated meat. Kakash et al. [86] also studied the effect of actinidin on the chicken thigh muscle and noted a positive effect on reducing the hardness and punch force and also on reducing the pH.

The enzyme extract of kiwifruit had a higher ability to hydrolyze commercially available substrates and beef connective tissue and myofibrillar proteins than the asparagus enzyme extract. Combining both these had a synergistic effect [91]. Pear protease has been recently explored as an ideal meat tenderizer with the desired meat tenderization level rather than overaction leading to mushy spots due to over proteolysis as with bromelain and actinidin [92]. In plant protease extracted from four overmatured fruits, viz., kiwifruit, pear, apple, and grape, kiwifruit demonstrated potent protease activity (921 U) followed by grape (225.86 U) > pear (97.75 U) > apple (78.29 U). The optimum temperature range for protease activity is 50–60 °C. A mixture of all these plant proteases from overmatured fruits had a good tenderization effect in beef sausage [93].

### 4.2. Zingibain

Zingibain (EC3.4.22.67) is a protease extracted from the ginger rhizome of *Zingiber officinale*. It has optimal enzymatic activity at 60 °C and pH 5.5 [94]. The maximum enzyme activity of zingibain was recorded at 60 °C and pH 7.0. It is stable at 40–65 °C temperature for 2 h, and is highly stable in the presence of metal ions. Zingibain extracted by three-phase partitioning had 215% recovery, 14.91 purification fold, 33.8 kDa molecular mass, and an isoelectric point at pH 4.38 [95].

Zingibain tends to break down collagen in meat compared to other plant proteases that break down the myofibrillar protein [95]. Sullivan et al. [79] reported that treating meat samples with homogenized fresh ginger paste resulted in an off flavor in the meat. Cruz et al. [94] studied the effect of crude ginger extract on the *Pectoralis major* muscle of broiler or chicken breast. The samples were injected with crude ginger extract, and the results showed decreased shear force and myofibrillar fragment index while increased myofibrillar fragment length without higher cooking loss. These factors ultimately resulted in improved tenderness of the chicken breast without degrading other qualities.

Proteases in the crude ginger extract obtained from ginger rhizome peel (homogenized in 100 mM phosphate buffer, pH 7, centrifugation at 10,500× *g* for 30 min at 4 °C) was reported to be stable at high temperatures with optimum activity at 60 °C at pH 5.5. Its application (5% *w*/*v*) to the pectoralis major muscle in the broiler breast caused increased tenderness by decreasing shear force (*p* < 0.05), increasing (*p* < 0.01) myofibrillar fragment index, and increasing (*p* < 0.001) myofibrillar fragment length [94]. The application of ginger extract from fresh ginger roots (5%) along with two commonly used protease enzymes (50 ppm bromelain, 50 ppm bromelain with 20 ppm papain) in the chicken breast muscle resulted in a significant increase in cooking loss and collagen solubility and markedly reduced thermal shrinkage treatment [96].

## 5. Novel Technological Interventions

There has been an increasing trend in utilizing exogenous plant proteinase to improve meat tenderness and eating qualities. The commercial production of plant protease is mainly used from five sources, viz., papain from papaya (*Carica papaya*), bromelain from pineapple (*Ananas comosus*), ficin from fig (*Ficus carica)*, actinidin from kiwifruit (*Actinidia Lindl*), and zingibain from the ginger rhizome (*Zingiber officinale*). The enzymatic activity is affected by the processing conditions such as time–temperature combinations, pH, and enzyme–surface contact surface.

There is an urgent need felt by the meat industry to regulate the enzymatic activity properly, enzyme diffusion in the meat matrix, and its overaction potential, so as to get the desirable effect on tenderness, texture, color, flavor, and juiciness [28,97]. Meat tenderization with plant proteases takes a longer time; thus, there are chances of the deterioration of meat color. Applying novel technologies such as high-pressure processing, ultrasonication, shock waves, hydrodynamic pressure, pulse electric field, etc., can improve the efficacy of enzymes by improving the enzyme–substrate ratio due to better penetration/diffusion of the enzyme inside the meat tissue. Various prospects for the application of novel processing technologies in the tenderization of meat by the application of plant proteases are described in Figure 3.

### 5.1. Ultrasound-Assisted Plant Proteases Tenderization of Meat

Ultrasound is a non-thermal, emerging green technology used for improving the freshness, cleanness, eating quality, and safety of food products with minimal heat loss and nutrient degradation [99]. Compared to traditional thermal technologies, ultrasound reduces processing time, saves labor due to automation, preserves nutritive and sensory attributes, and increases the shelf life of foods [100,101]. It is applied for drying, cleaning, tenderization, sterilization, thawing, freeze-drying, extraction, changing the structure and properties of food proteins, improving substrate solubility, and curing in the food industry [102,103]. Its tenderization effect in meat is due to the cavitation effect and physical shearing leading to the destruction of the myofibrils structure [104]. It also increases the accessibility of enzymes by increasing the penetration of enzymes resulting in an improved tenderization effect [105].

Ultrasonication of meat has an impact on various meat microstructures, meat color, tenderization, pH, juiciness, and water-holding capacity [106,107]. In the food industry, high-frequency (>1 MHz) and low-intensity (<1 W/cm^2^) ultrasound radiations are used for the characterization of chemical and physical attributes of foods such as hardness, acidity, and maturity, whereas low-frequency (20–100 kHz) and high-intensity (10–1000 W/cm^2^) are applied to assess the impact of physical or chemical processes in the food sector [99,104,106,107].

The application of ultrasound waves (ultrasound probe 20 kHz, 100 W power for 20 min) in combination with papain (0.1% papain) was observed to have significantly (*p* < 0.05) improved tenderness and proteolytic activity as indicated by decreased filtering residue, shear force, and textural profile in *longissimus lumborum* muscles of young Holstein bulls [103]. The combined treatment of beef caused microstructural changes/damages that facilitated papain diffusion into the deeper part of the beef, leading to improved tenderness and flavor [102].

The ultrasound-assisted papain treatment (Ultrasound—300 W, 40 kHz, 30 °C for 20 min; papain—45 U/g injecting to 1/10 sample weight) to spent hen meat was observed in improving the eating quality of spent hen meat (shear force, WHC), reduced drip loss, improved water retention by promoting the migration and uniform distribution of water in meat, and reducing the meat color score and pH. Ultrasound treatment of more than 20 min was observed to cause a reduction in shear force and WHC [104]. Microstructure analysis revealed the destruction of muscle fiber structure. Cao et al. [104] also observed ultrasound radiation as a promising auxiliary method that can improve meat quality synergistically.

### 5.2. High-Pressure Processing Assisted Plant Proteases Meat Tenderization

High-pressure processing (HPP) is a novel green, non-thermal technology used to improve food products’ microbial safety by destroying microorganisms in food and food quality [28]. HPP treatment improves the tenderization of meat by increasing protein solubility due to the depolymerization/unfolding of proteins under high pressure, preferably at high temperatures. Ground chicken meat treated with HPP at 350 MPa for 4–12 min at 4 °C resulted in a 1.5–3.5 log reduction of *Salmonella* spp. [108]. An injection of papain and isoelectric pressure of 100 MPa for 10 min resulted in significantly improving the tenderness of the *longissimus lumborum* muscle of beef, with no significant changes in meat tenderness but a higher pale color recorded on further increasing the pressure [109]. However, upon cooking at 65 °C for 40 min, the meat sample treated with papain alone or with pressure-treated samples had the same level of color development [109].

The application of papain (80 U/mL, 2 h, 55 °C) to yak meat, however, improved water holding capacity (9.9%) and increased the tenderness score of meat by decreasing shear force (46.9%), but negatively affected the color score [110]. The authors also observed that a combined application of papain (80 U/mL, 30 min, 55 °C) followed by HPP (50 MPa, 15 min) had a synergistic effect and decreased shear force more than the single treatment without compromising the color of the beef.

Li et al. [77] reported that 0.10 g/L of ficin reduced the hardness of tan mutton by 8% while preserving the color of the meat. However, the concentration of ficin that was applied also reduced the WHC, resulting in an increased centrifugal loss or water loss of the meat. Nonetheless, this situation can still be controlled and improved by incubating the treated sample with 0.10 g/L ficin at 55 °C for 1 h, followed by high pressure at 50 MPa or 150 Mpa for 15 min. This treatment sequence could reduce the meat hardness by 48% if 50 Mpa of pressure was used and 41% if 150 Mpa of pressure was applied. Other than that, this treatment can also reduce cooking loss and centrifugal loss [77].

Shockwaves or hydrodynamic pressure processing (HDP) generates instantaneous high pressure in water through shock waves by electric discharge, and this pressure can be transferred via a wave of pressure in water to any material that has an acoustic match for water such as meat with about 75% water [111].

### 5.3. Immobilization

Immobilization of enzymes improves the solubility of enzymes in an aqueous medium, recycling enzymes, improving the activity and stability of enzymes by protecting enzymes from intermolecular phenomenon, and increasing the rigidity of the enzyme structure by multipoint covalent attachment, thereby minimizing conformational changes [15,112].

The limited shelf-life of plant proteases proves a technological challenge for the meat industry. The incorporation of plant proteases into edible films (such as 5–10% papain incorporation into cassava starch edible films for beef packaging) had the advantages of controlled action, improved tenderization, and preventing excessive activity leading to an indiscriminate breakdown of myofibrillar tissues, connective tissue, and collagen [113]. The low shelf-life due to the poor stability of plant proteases could be improved by the adsorption of proteases on chitosan, later being used for packaging applications in the food industry.

Immobilization of cysteine proteases results in marked deterioration of enzyme activity on adsorption on chitosan such as sorption of 49–64% for bromelain, 34–28% for papain, and 69–70% for ficin upon adsorption on medium to high molecular weight chitosan, respectively [114]. The authors [114] observed 5–19% destruction of the helical structure of protein and sorption of proteases on chitosan by L and R domains including active sites of enzymes. This adsorption was not observed to affect the optimum pH range and temperature for enzymatic action, however, it significantly improved the stability of proteases with 5.8 times higher stability for bromelain and 7.6 times higher stability for papain [114].

The immobilization of actinidin by using gold nanorods could improve the stability and activity of actinidin. Homaei et al. [115] observed improved actinidin release at 40–60 °C and pH 7–8.5 upon immobilization on gold nanorods by ionic exchange and hydrophobic interaction. The process has resulted in markedly improving the shelf life and stability as well as significantly improving resistance against the inhibitory effects of various bivalent metal ions. The authors [115] reported higher functionality for actinidin in immobilized form as compared to free form.

Bromelain has the potential to effectively enhance the utilization of low-value/underused cuts of meat by using restructuring technologies. Shin et al. [116] applied double emulsion for controlled release and improving bromelain (1% *w*/*v*) stability in the tenderization of pork loin and observed significantly (*p* < 0.05) higher water holding capacity, lower (*p* < 0.05) cooking loss, and degradation of the myosin heavy chain. The application of bromelain (0.05–0.1% *w*/*w* at 50 °C for 12 min) in the preparation of restructured pork stew resulted in decreasing shear force value and texture profile analysis attributes but higher total protein, sarcoplasmic protein solubility, total collagen, and soluble collagen [117].

### 5.4. Combinations of Proteases

Various plant proteases have different types of action by having various sites of action in meat, thus applying a combination of plant proteases could be a good strategy for getting the desired degree of tenderization. Similarly, optimum tenderization of semitendinosus muscle containing collagen and elastin could be achieved by plant proteases such as bromelain, papain, and ficin [118]. Bromelain degrades the collagen more whereas ficin has a more balanced degradation of both myofibrillar and collagen proteins [79].

Among plant protease, papain, bromelain, and ficin at 100 ppm each had a tenderizing effect (lower hardness and shear force value) on the adductor muscle of camel, with the enzymatic efficiency in the following order of decreasing papain > bromelain > ficin [81]. A combination of papain and bromelain (210 U/100 mL, pH 6.4, 30 °C for 40 min) has been used for tenderizing the tough and rigid mantle of jumbo squid mantle (*Dosidicus gigas*), so to develop an acceptable product for consumers [119]. The authors [119] also reported a significant decrease (*p* < 0.05) in shear force, muscle hardness, myofibrillar protein content, and Ca^2+^ ATPase activity. The enzymatic treatment had broken myofibrils and produced a large number of small fragments in the muscle tissues, thus decreasing microstructure stability and integrity [119].

Kim et al. [120] applied papain, bromelain, and other animal proteases (pepsin and pancreatin) to tenderize the hard texture of horse meat (thigh muscle of horse, 50–55 °C for 1–8 h duration in a water bath with enzymes), so to make it acceptable to elder people having difficulty in mastication and dental problems. The authors [120] observed significant improvement in tenderness by significantly (*p* < 0.001) reducing hardness and resilience, thus making it suitable for elderly people with teeth problems/ mastication difficulties.

Nam et al. [92] explored the potential to use purified pear protease as a meat tenderizer and noted that pear protease and kiwi fruit protease have synergistic effects and could produce good quality meat products. Pear protease was stable at a wide pH range (5–8) with a catalytic efficiency of 2.9–2.7 μM/min. Pear protease stabilized in 5% dextrin during lyophilization demonstrated good proteolytic activity on casein and beef myofibrillar proteins [92]. A combination of the proteolytic enzyme of kiwifruit and pineapple and sous vide cooking was observed to demonstrate a significant reduction in the hardness of pork foreshank, and an increase in the L* and b* color value with kiwifruit enzyme was 60% more effective than pineapple protease and commercial neutrase [121].

By-products of the pineapple and jackfruit industry comprising core and seeds were observed to have a potent tenderizing effect in beef [122]. There was a significant decrease (*p* < 0.05) in cooking yield, water holding capacity, moisture content, and improving sensory attributes of beef upon incorporation with increased levels of crude enzymes from 0–4%. Several gaps or spaces in the beef muscle treated with crude proteases were observed upon microstructural analysis [122]. The incorporation of pineapple rind and fig powder (1.5% each powder) was observed to have a tenderizing effect on the development of restructured spent hen meat slices [123].

The application of various plant proteases in meat tenderization is presented in Table 2.

## 6. Novel Source of Plant Protease

Plant proteases/proteinases can be harvested from vast untapped plant resources with suitable extraction technologies. Some plant powders are traditionally added during the preparation of meat products to improve their sensory and nutritional quality. Plant protease extracted from cashew (*Anacardium occidentale*) fruit was reported to have a tenderizing effect in the meat [33]. The authors [33] extracted protease (activity-6.302 U/mL) from the cashew apple or the flesh of the cashew fruit by blending the cashew apple with a chilled buffer solution. The crude extract of the cashew apple was filtered using cotton clothes. The centrifugation process was conducted for 20 min at 4 °C and the supernatant was stored under refrigeration conditions to remove remaining undigested particles. The extracted crude protease (9%) application to the beef tenderloin for 24 h at 4 °C resulted in decreased shear force, complete protein band degradation, and disruption of the muscle fibers. These results show the positive effect of protease from cashew fruits in tenderizing meat [33].

Proteases extracted from cassava (*Manihot esculenta*) root flesh by the centrifugation of pulp in a buffer solution at pH 9, 6.32 mM 2-mercaptoethanol, 4.12% Triton X-100, 3.24 mM Calcium chloride, 9910 rpm for 30 min [126]. The beef samples treated with the crude extract of *Manihot esculenta* showed a reduction of shear force with the application of 0.9 mL crude *Manihot esculenta* root extracts resulting in the lowest firmness by extensively damaging the protein bands, degradation of myofibrillar proteins, and connective tissues.

A novel protease obtained from wild cantaloupe (*Cucumis trigonus* Rox-b) fruit by drying the peels in the oven at 37 °C was observed to have a meat tenderizing effect [12]. The samples were immersed in the enzyme extract solution at different concentrations and then were taken out, washed, and drained. Next, the samples were stored in tightly sealed polyethylene bags and kept at 4 °C for 48 h before being transferred to a freezer at −18 °C. The treatment resulted in a decrease in pH, collagen content, water-holding capacity (WHC), and increased protein solubility [127].

The proteases extracted from ambarella (*Spondias cytherea*) fruit pulp were reported to be stable at pH 8.0–10.0 and a temperature ranging from 50–60 °C [128]. The pulp was chopped into small pieces and then blended in the mixer with chilled buffer solution (50 mM phosphate buffer, 50 mM tris(hydroxymethyl)aminomethane (Tris) buffer, and 50 mM glycine–NaOH buffer) for 3 min. The crude extract was filtered and centrifuged at 15,000 g for 20 min at 4 °C. The crude proteases extract was applied to beef samples at 0.3, 0.6, and 0.9 mL for a constant marination period and temperature of 24 h and 4 °C, respectively. The textural analysis results have shown a reduction in meat firmness and shear force for all the treated samples compared to the untreated sample [128]. A marked loss of enzymatic activity (up to 32%) in this protease extracted from ambarella was also reported by authors upon prolonged storage [128].

A 5% aqueous extract of *Sarcodon aspratus* mushroom has been demonstrated to exert a tenderizing effect on the *longissimus thoracic et lumborum* muscle of cattle by degrading actin and heavy chain myosin resulting in improving WHC and improved sensory attributes [129]. The crude enzyme extract (0.5% *w*/*w*) obtained from the latex of giant milkweed or small crown flower (*Calotropis procera*) demonstrated a tenderizing effect in pork, beef, and chicken upon marination for 1 h at 4 °C [130]. The protease treatment of meat resulted in a significant (*p* < 0.05) decreased toughness, an increase in protein solubility, and the sign of extensive proteolysis upon electrophoresis [130].

The crude extract of the mango (*Magnifera indica* L.) peel was prepared by cutting the mango peel into small pieces and then blended with a 0.1 M sodium phosphate buffer with pH 7.5 at 4 °C for 2 min. After that, the blended peel was filtered using a cheese cloth followed by centrifugation for 15 min at 4 °C [131]. The mango peel crude extract was applied to the beef sample from the top round and kept in a Ziplock bag at 25 °C with three different marination times, viz., 1 h, 2 h, and 3 h. The authors observed a reduction in shear force, improved collagen solubility, and degradation of the myofibrillar protein and connective tissues leading to improved beef tenderness [131].

The soup containing jackfruit (*Artocarpus heterophyllus*) leaf powder was observed to improve meat tenderness [132]. The meat samples marinated for 1 h before being cooked in the oven for 8 min showed the decreased hardness of the treated samples. The authors [132] observed that jackfruit leaf had the protease that could benefit meat tenderness and incorporating jackfruit leaf powder into the soup spice powder had the potential to be commercialized [132].

The research on new sources of plant proteases and the effect on meat tenderness and other effects is summarized in Table 3.

## 7. Prospects and Challenges

The meat industry increasingly prefers the application of plant proteases in meat tenderization due to their easier availability, lack of food safety risks, acceptability to all sections of consumers, enzymatic potential, and absence of ethical and animal welfare issues. The application of combinations of plant proteases has been reported to exert synergistic effects, and it is thus easier to get a desirable product quality for even very tough and old meat. The issue of immobilization or stabilization of these novel plant proteases should be increased, so as to make the tenderization process more economical and sustainable.

The optimum and controlled tenderization of meat by plant protease will help to decrease the batch-wise variability in the meat and meat products and improve consumer acceptability. Significantly reducing the hardness of meat by plant protease makes these products suitable for consumption by old-age people (having reduced mastication and chewing capacity) required for fulfilling balanced nutrition and supply of high-quality proteins [133].

The extraction of plant proteases from vast natural resources, fruit industry by-products, and wastes has the potential to guarantee the regular supply of these exogenous plant proteases in the future. The plant protease concentration is varied with the climate, soil, plant part, age of plants, soil quality, harvesting time, and post-harvesting processing conditions. Genetic engineering, suitable plant breeding practices, and cell culturing could increase the enzyme content in plant biomass. There should be a focus on the introduction of novel, green, efficient technology for improved extraction of the enzyme from plant biomass and their separation and purification while preserving enzymatic activity.

The application of novel meat tenderizing processing technologies such as ultrasound radiation, high-pressure processing, reverse phase micellar, shock waves, and pulse electric field are increasingly used to improve plant protease efficiency and reactivity by deep penetration and improve the efficiency and reactivity of plant proteases by a deep penetration of the enzyme–substrate ratio. To get better and more desirable meat quality, a combination of two or more plant proteases is applied during meat tenderization. The application of more than one protease has a synergistic effect.

The proteolytic action of exogenous plant proteases on muscle protein increases the nutritive value of food by increasing the amount and availability of essential amino acids, improving digestibility and palatability, as well as the production of bioactive peptides possessing health benefits [7,134,135]. The application of bioinformatics by using an in silico approach and an in vivo approach could be used for the rapid identification of bioactive peptides generated during the tenderization of meat by exogenous plant protease and could lead to predictive bioactivity and peptidomics [7,136,137]. The action of plant proteases has the potential to improve the survival rate of bioactive peptides during digestion. Similarly, actinidin treatment increased the in vitro digestibility of meat proteins [83], and cooked meat proteins under simulated gastrointestinal digestion [138] and the produced angiotensin I-converting enzymes (ACE) inhibitory peptides [2] with less than 3 kDa fraction had the highest ACE inhibitory activity leading to potential anti-hypertension applications [139].

Plant proteases have proven to be effective as a meat tenderizer. However, certain parameters need to be controlled, which include the enzyme concentration, pH, treatment time, and treatment temperature for the proteases to work optimally. In some cases, the treatment with the proteases alone is not favorable as it can degrade the meat quality in terms of color, flavor, moisture content of meat, cooking loss, and WHC. Thus, a combination with other mechanical treatments such as HPP is needed. Over tenderization of meat by plant proteases in uncontrolled conditions leading to musty or decomposed meat still remains a challenge for the meat industry. It impacts the flavor, texture, and consumer acceptability of meat products. Excessive proteolysis also impacts taste by increasing bitterness due to the formation of basic amines and amino acids.

The issues of wide variabilities in the action of exogenous enzymes, selectivity towards connective tissue versus myofibrillar proteins, and the overall impact on meat tenderness are some critical areas that need to be addressed to apply plant proteases in the meat industry. There is a need to undertake further research to establish food safety with the application of a novel source of plant proteases, such as kiwi fruit protease causing allergic reactions in some individuals. However, further research on the effects of the novel proteases on other important parameters such as color, sensory experience, the possibility of causing hypersensitivity, and toxicity is needed.

## 8. Conclusions

Plant proteases are efficient meat tenderizers as they reduce shear force value and degrade myofibrillar proteins and connective tissue. The current focus of the meat industry during the tenderization process remains on increasing nutritive value, production of bioactive peptides, tender-optimized meat products for targeted populations, and the production of modified and uniform texturized meat products. Reusing the plant proteases, improving survival rate, and harvesting novel plant proteases with desirable kinetics and thermodynamics for improving circular economy and reducing food waste are some recent areas in the application of plant proteases in meat tenderization.

There are limited studies on the nutritional changes in meat treated with plant proteases. Improving digestibility, increasing nutritive value, and generating bioactive peptides with improved survival rates are recent trends observed in the current applications of plant proteases in meat. Future research is needed to assess the efficiency of the proteases obtained from novel plant sources in improving meat tenderness.

## Figures and Tables

**Figure 1 foods-12-01336-f001:**
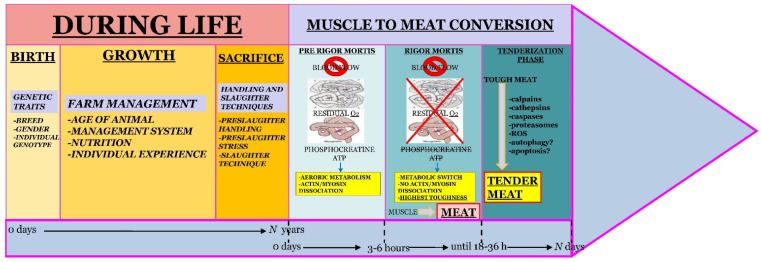
Timeline for development of meat tenderness. [Note: After rigor mortis, in the tenderization phase plant proteases are added to improve the meat tenderization by degrading connective tissue and myofibrillar proteins]. “Reprinted from Ref. [25]. Copyright (2016) with permission from Elsevier”.

**Figure 2 foods-12-01336-f002:**
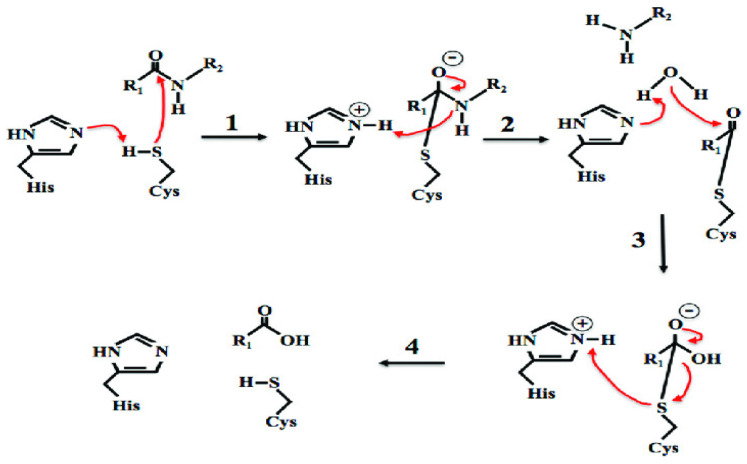
The classic cysteine protease reaction mechanism. Adopted from [44].

**Figure 3 foods-12-01336-f003:**
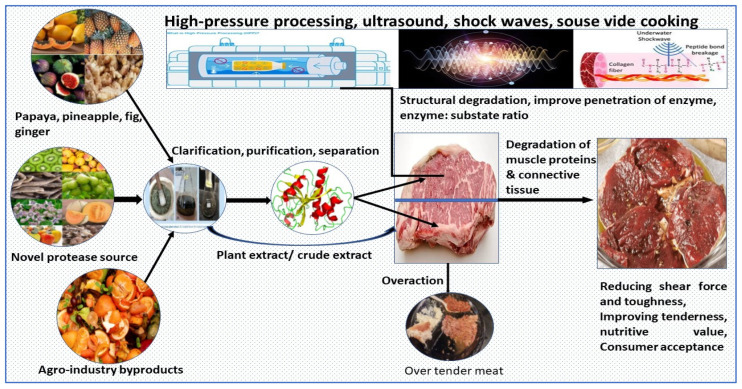
Various prospects of the application of novel processing technologies in the tenderization of meat using the application of plant proteases. (Source of underwater shockwaves—Fan et al. [98]).

**Table 1 foods-12-01336-t001:** Physiological attributes of some common plant proteases.

Name of Enzymes	Enzyme Commission Source	Source	pH Range (Optimal Range)	Temperature (°C, Optimal Range)	Principal Action Site
Papain	EC 3.4.22.2	Papaya latex	4.0–9.0 (6.0–7.0)	40–80 (60–75)	Myofibrillar protein, collagen
Bromelain	EC 3.4.22.32	Stem of pineapple	5.0–8.5 (6.0–8.5)	50–80 (50–65)	Collagen, myofibrillar protein
Ficin	EC 3.4.22.3	Fig latex	4.0–9.0 (5.5–7.5)	40–70 (45–60)	Collagen, myofibrillar protein
Actinidin	EC 3.4.22.14	Kiwi fruit	5.0–8.0 (7.0–8.5)	40–60	Myofibrillar proteins
Zingibain	EC 3.4.22.67	Ginger rhizome	5.0–8.5 (6.0–7.0)	40–70 (60–70)	Collagen, myofibrillar protein
Cucumin	-	*Cucumis trigonus* Roxb	5.0	40–70	Collagen, myofibrillar protein

Sources [5,7,40,41,42,43].

**Table 2 foods-12-01336-t002:** Application of plant proteases in meat tenderization.

Enzyme	Sample	Treatment	Positive Effect	Negative Effect	Reference
Papain	Spent hen leg cuts	- 0.025% papain - Infusion and forking technique	- Improve tenderness, salt soluble protein, WHC, emulsifying tendency, and emulsion stability	N/A	[124]
Papain	Yak meat	- Injection of 80 U/mL papain	- Reduce shear force - Improve tenderness - Increase WHC	- Degrade color	[110]
		- Injection of 80 U/mL papain - Incubation for 30 min at 55 °C - Followed by HPP at 50 MPa for 15 min	- No color changes - Improve tenderness - Increase WHC	N/A	
Papain	Jumbo squid (*Dosidicus gigas*)	- Incubation with papain solution for 40 min in a 30 °C water bath	- Improve tenderness - Reduce the muscle hardness - Reduce shear force - Reduce myofibrillar protein content	- Reduce WHC	[119]
Bromelain	Brahman cattle round cuts	- Treatment with freeze-dried bromelain powder extracted from pineapple crown	- Decrease hardness	- Reduce WHC - Reduce moisture content - Degrade color - Increase cooking loss	[65]
Bromelain	Buffalo meat	- Soaking with pineapple fruit extract	- Improve tenderness - Increase (WHC) - Reduce the cooking loss or shrinkage	N/A	[62]
Bromelain	Beef round cuts	- Treatment with bromelain powder from pineapple crown	- Increase amino acids content - Increase tenderness	- Reduce WHC	[70]
Ficin	Camel round cuts	- Treatment with 100 ppm papain, stored at a temperature 4 °C for 4 days	- Decrease hardness, springiness, cohesiveness, gumminess, chewiness, and shear force - Increase tenderness	N/A	[79]
Ficin	Mutton	- Treatment with 0.10 g/L ficin	- Reduce the hardness of tan mutton by 8% - Preserving the color of meat	- Reduce WHC	[77]
		- Incubation of the treated sample with 0.10 g/L ficin at 55 °C for 1 h and followed by HPP at 50 MPa or 150 MPa for 15 min	- Reduce the meat hardness by 48% if 50 MPa of pressure used and 41% if 150 MPa of pressure used - Reduce the cooking loss and centrifugal loss - High sensory score for tenderness, flavor, and texture	N/A	
Actinidin	Pork muscle and	- Treatment with actinidin at 0.5 mg/100 g of muscles	- Increase tenderness - Reduce shear force	N/A	[2]
	rabbit muscle				
Actinidin	Brisket steaks	- Injection of 5% of a 3 mg/mL solution of commercial actinidin extract (Actazin™ from Anagenix Ltd.) - Followed by vacuum tumbling and sous vide cooking process for 30 min at 70 °C	- High acceptability - High sensory score for tenderness, juiciness, and flavor - No changes in pH, color, and cooking loss	N/A	[83]
Zingibain	Chicken breast	- Treatment with crude ginger extract	- Reduce shear force - Increase myofibrillar fragment index - Reduce myofibrillar fragment length - Increase tenderness - Not affecting cooking loss	N/A	[94]
Cucumin	Buffalo meat	Cucumis extract (5% *w*/*v*) and ginger extract (0.2% *w*/*w*)	- Increased collagen and myofibrillar protein solubility - Lowered shear force - Improved sensory attributes	- Extensive proteolysis	[43]
Cucumin	Goat meat curry	Cucumis powder (2.0%) spray, citric acid (1.0%)	- Increased soluble collagen, lowered shear force	-	[125]

**Table 3 foods-12-01336-t003:** Novel sources of plant protease for meat tenderization.

Source	Meat Sample	Extraction Method	Treatment	Result	Reference
Cashew fruit (*Anacardium occidentale*)	Beef (Tenderloin)	Centrifugation of crude cashew flesh extracts for 20 min at 4 °C	- Spraying different concentrations of crude protease extract on the samples - Marinated for 24 h at 4 °C	- Treatment with 9% crude protease extract has the lowest shear force, complete protein bands degradation, and disruption of muscle fibers of beef tenderloin	[33]
*Manihot esculenta*	Beef	Centrifugation of crude flesh *Manihot esculenta* extract at 9910 rpm for 30 min	- Marination of samples with different volumes of crude protease extract	- Treatment with 0.9 mL crude protease extract resulted in the lowest firmness of beef	[126]
Wild cantaloupe fruit (*Cucumis trigonus* Rox-b)	Fresh, aged bull meat (Aged for more than 5 years)	The cantaloupe peel was dried, ground, and transformed into a solution	- The meat samples were immersed in different concentrations of dried cantaloupe peel solution and were taken out, washed, and drained. - Samples were stored in tightly sealed polyethylene bags, kept at 4 °C for 48 h, and then transferred to a freezer at −18 °C.	- pH decreased - Collagen content decreased - Protein solubility increased - WHC decreased	[127]
Nam Dok Mai Mango (*Magnifera indica* L. var. *Nam Dokmai*)	Beef (Top round cuts)	Centrifugation of crude mango peel extract for 15 min at 4 °C	- Fork-stabbed samples marinated with crude protease extract for different durations	- Reduction of shear force, collagen solubility, protein extractability, and TCA-soluble peptide content - Degradation of the myofibrillar protein and connective tissues	[131]
Ambarella (*Spondias cytherea*)	Beef	Centrifugation of crude Ambarella pulp extract for 15,000× *g* for 20 min at 4 °C	- -Marination with different volumes of crude protease extract for 24 h at 4 °C	- Reduction in meat firmness and shear force - Muscle fibers of samples treated with Ambarella protease were degraded	[128]
Jackfruit (*Artocarpus heterophyllus)*	Beef (Bottom round cuts)	The jackfruit leaf was dried, ground and mixed with other spices to produce soup spice powder	- Marination of samples with jackfruit leaf soup spice powder for 1 h	- Hardness of the treated samples was decreased compared to the untreated sample	[132]

## Data Availability

No new data were created.

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
