# Peer review of "Application of Plant Proteases in Meat Tenderization: Recent Trends and Future Prospects"

_foods, 2023, doi:10.3390/foods12061336_

Round 1

Reviewer 1 Report

The review “Application of Plant Proteinase in Meat Tenderization: Recent Trends and Future Prospects” is devoted to an actual topic and has a high applied value.

It is known, that plant proteases have been used for a long time as active ingredients in tenderization of meat in the food industry. The authors emphasized in the article, that the uncontrolled plant protease action could cause over-tenderization and poor quality. The present review critically analyzed recent trends and future prospects of the application of plant proteases in meat tenderization.

The structure of the review is logical and consistent.

I have a small comment: in Section 3, the authors should add information on the kinetic (Km, Vmax, kcat) and thermodynamic properties of plant proteases.

Minor remarks:

Lines 31-34: “The meat industry is targeting to reuse of enzymes and improving their stability and shelf-life by immobilization, encapsulation, protein engineering, medium engineering, immobilization, and stabilization of plant proteases during tenderization” – the immobilization word occurs twice.

Lines 43-44: text broken.

Line 282: in the word “N-α-CBZ-lysine p-nitrophenyl ester” N and p should be italicized.

References starting from number 100 come with a paragraph gap, which should be removed.

Author Response

The authors are thankful to the anonymous reviewer for his critical comments and observations. We have edited the manuscript accordingly. Further, it is certified that all the issues raised by the reviewer have been incorporated into the revised manuscript. All the changes were marked in RED color text.

A detailed description of reviewer comments and author responses are as follows-

Comment: It is known, that plant proteases have been used for a long time as active ingredients in the tenderization of meat in the food industry. The authors emphasized in the article, that the uncontrolled plant. protease action could cause over-tenderization and poor quality. The present review critically analyzed recent trends and future prospects of the application of plant proteases in meat tenderization

Response: Authors thank the reviewer for positive and encouraging observations.

Comment: The structure of the review is logical and consistent.

Response: Authors thank the reviewer for positive and encouraging observations.

Comment: I have a small comment: in Section 3, the authors should add information on the kinetic (Km, Vmax, kcat) and thermodynamic properties of plant proteases.

Response: Thank you so much for your valuable suggestions. A section on kinetics and thermodynamics has been added in section 3.

Comment: Lines 31-34: “The meat industry is targeting to reuse of enzymes and improving their stability and shelf-life by immobilization, encapsulation, protein engineering, medium engineering, immobilization, and stabilization of plant proteases during tenderization” – the immobilization word occurs twice.

Response: The repeated word deleted.

Comment: Lines 43-44: text broken.

Response: Corrected

Comment: Line 282: in the word “N-α-CBZ-lysine p-nitrophenyl ester” N and p should be italicized.

Response: Edited

Comment: References starting from number 100 come with a paragraphgh, which should be removed.

Response: Edited

Reviewer 2 Report

REVIEW Application of Plant Proteinase in Meat Tenderization: Recent Trends and Future Prospects

 In the tile appears "proteinase" but in the main text "protease". I recommend use the same word in title and text.

The numbers assigned to sub-titles are not correct, because there are a lot of number 3

 Previously, some reviews showed information about plant proteinase in meat tenderization, thus it would be interesting to mention these antecedents in the introduction section and highlight what is the novelty that this manuscript provides.

Here some reviews published:

Bekhit, A. A., Hopkins, D. L., Geesink, G., Bekhit, A. A., & Franks, P. (2014). Exogenous proteases for meat tenderization. Critical reviews in food science and nutrition54(8), 1012-1031.

Bolumar, T., Enneking, M., Toepfl, S., & Heinz, V. (2013). New developments in shockwave technology intended for meat tenderization: Opportunities and challenges. A review. Meat Science95(4), 931-939.

Fernández-Lucas, J., Castañeda, D., & Hormigo, D. (2017). New trends for a classical enzyme: Papain, a biotechnological success story in the food industry. Trends in Food Science & Technology68, 91-101.

Gagaoua, M., Dib, A. L., Lakhdara, N., Lamri, M., BotineÅŸtean, C., & Lorenzo, J. M. (2021). Artificial meat tenderization using plant cysteine proteases. Current Opinion in Food Science38, 177-188.

Jabalia, N., Mishra, P. C., & Chaudhary, N. (2014). Applications, challenges and future prospects of proteases: An overview. Journal of Agroecology and Natural Resource Management1(3), 179-183.

Madhusankha, G. D. M. P., & Thilakarathna, R. C. N. (2021). Meat tenderization mechanism and the impact of plant exogenous proteases: A review. Arabian Journal of Chemistry14(2), 102967.

Morellon-Sterling, R., El-Siar, H., Tavano, O. L., Berenguer-Murcia, Á., & Fernández-Lafuente, R. (2020). Ficin: A protease extract with relevance in biotechnology and biocatalysis. International journal of biological macromolecules162, 394-404.

Shi, H., Shahidi, F., Wang, J., Huang, Y., Zou, Y., Xu, W., & Wang, D. (2021). Techniques for postmortem tenderisation in meat processing: effectiveness, application and possible mechanisms. Food Production, Processing and Nutrition3, 1-26.

Singh, R., Singh, A., & Sachan, S. (2019). Enzymes used in the food industry: Friends or foes?. In Enzymes in food biotechnology (pp. 827-843). Academic Press.

Verma, V., Singhal, G., Joshi, S., Choudhary, M., & Srivastava, N. (2022). Plant extracts as enzymes. In Plant Extracts: Applications in the Food Industry (pp. 209-223). Academic Press.

Warner, R. D., Wheeler, T. L., Ha, M., Li, X., Bekhit, A. E. D., Morton, J., ... & Zhang, W. (2022). Meat tenderness: Advances in biology, biochemistry, molecular mechanisms and new technologies. Meat Science185, 108657.

Section “Meat tenderization”. The last paragraph (Proteases are present in all plants since being vital compounds needed for the plant's physiological and development process [16]. Rec ently, the application of exogenous en zymes for meat tenderization has become an increasing focus of interest among food technologists and meat scientists [17]. However, there is a need to undertake comprehensive  studies on the safety and efficiency of meat tenderizers and find out which plant sources  are suitable for the meat tenderizer. The food industry is mainly focused on obtaining  plant proteinase from commercial raw materials such as papaya, fig, pineapple, and microorganisms. The extraction, identification, and purification of plant proteinase from  novel non-conventional sources can significantly contribute to the future supply of commercial exogenous plant proteinase in the near future.) should be in section “Plant protease as a natural meat tenderizer”

Section “Plant protease as a natural meat tenderizer”. Plant enzymes are also called proteases, proteinases, and peptidases [18]. This terms are not exclusive for plants, there are used in other organism too.

Previously to figure 1, should be necessary to include some information about cysteine proteinases.

Tabla 1 should be appears before Figure 1, where the enzymes bromelain, papain, ficin, actinidin, and zingibain are mentioned.

In the sections “Papain extraction and application”; “Bromelain application and extraction”; “Ficin application and extraction”, perhaps is convenient include the enzymes names, without extraction and application; only Papain; bromelain and Ficin in titles.

Section “Miscellaneous plant proteases”. Should be interesting to add a introductory paragraph about the enzymes mentioned in this subtitle

Section “Plant extracts as a meat tenderizer” is not necessary. This section should be eliminated and the information should go to other sections.

Paragraph 1- From “Plant extracts” to “meat products”. Send to “Plant protease as a natural meat tenderizer”

Paragraph 2- From “Proteases in the crude ginger extract” to “thermal shrinkage”. Send to “Zingibain”

Paragraph 3- From “Papaya extract  ” to “shrinkage after cooking”. Send to “Papain extraction and application”

Paragraph 4- From “Enzyme extract of kiwifruit” to “beef sausage  ”. Send to “Actinidin”

Paragraph 5- From “Nam et al explored” to “commercial neutrase”. Send to “Combinations of proteases”

Paragraph 6- From “The crude extract of the mango” to “improved beef tenderness”. Send to “Novel source of plant protease”

Paragraph 7- From “The soup containing jackfruit (Actocarpus…)” to “potential to be commercialized”. Send to “Novel source of plant protease”

Paragraph 8- From “By-products of the pineapple  ” to “meat slices”. Send to “Combinations of proteases”

Table 3 Send to “Novel source of plant protease”

Paragraph 9- From “Thus, plant proteases” to “other quality optimally”. Send to “Plant protease as a natural meat tenderizer”

Author Response

The authors are thankful to the anonymous reviewer for his critical comments and valuable suggestion for formatting the manuscript. These comments have helped us in improving the quality of the manuscript. Further, it is certified that all the issues raised by the reviewer have been incorporated into the revised manuscript. All the changes were marked in RED color text.

A detailed description of reviewer comments and author responses are as follow-

Comment: In the tile appears "proteinase" but in the main text "protease". I recommend use the same word in title and text.

Response: Edited and word proteases used as per text.

Comment: The numbers assigned to sub-titles are not correct, because there are a lot of number 3

Response: Formatted.

Comment: Previously, some reviews showed information about plant proteinase in meat tenderization, thus it would be interesting to mention these antecedents in the introduction section and highlight what is the novelty that this manuscript provides.

Response: Added the previous work and novelty (L62-64).

Comment: Section “Meat tenderization”. The last paragraph (Proteases are present in all plants since being vital compounds needed for the plant's physiological and development process [16]. Recently, the application of exogenous enzymes for meat tenderization has become an increasing focus of interest among food technologists and meat scientists [17]. However, there is a need to undertake comprehensive studies on the safety and efficiency of meat tenderizers and find out which plant sources are suitable for the meat tenderizer. The food industry is mainly focused on obtaining plant proteinase from commercial raw materials such as papaya, fig, pineapple, and microorganisms.The extraction, identification, and purification of plant proteinase from novel non-conventional sources can significantly contribute to the future supply of commercial exogenous plant proteinase in the near future.) should be in section “Plant protease as a natural meat tenderizer”

Response: Thank you so much for your valuable observation. We have shifted it.

Comment: Section “Plant protease as a natural meat tenderizer”. Plant enzymes are also called proteases, proteinases, and peptidases [18]. This terms are not exclusive for plants, there are used in other organism too.

Response: Deleted

Comment: Previously to figure 1, should be necessary to include some information about cysteine proteinases.

Response: Information on cysteine proteinases have been added.

Comment: Table 1 should be appears before Figure 1, where the enzymes bromelain, papain, ficin, actinidin, and zingibain are mentioned.

Response: Table shifted before fig as per valuable suggestion by the reviewer.

Comment: In the sections “Papain extraction and application”; “Bromelain application and extraction”; “Ficin application and extraction”, perhaps is convenient include the enzymes names, without extraction and application; only Papain; bromelain and Ficin in titles.

Response: Thank you for the suggestion, we have edited accordingly.

Comment: Section “Miscellaneous plant proteases”. Should be interesting to add a introductory paragraph about the enzymes mentioned in this subtitle

Response: Added

Comment: Section “Plant extracts as a meat tenderizer” is not necessary. This section should be eliminated and the information should go to other sections.

Response: The section deleted and content shifted to other sections as per suggestions.

Comment: Paragraph 1- From “Plant extracts” to “meat products”. Send to “Plant protease as a natural meat tenderizer”

Response: Para 1 shifted to section plant proteases as natural meat tenderizers.

Comment: Paragraph 2- From “Proteases in the crude ginger extract” to“thermal shrinkage”. Send to “Zingibain”

Response:Para 2 shifted to Papain section.

Comment: Paragraph 3- From “Papaya extract ” to “shrinkage after cooking”. Send to “Papain extraction and application”

Response: Para 3 shifted to Papain.

Comment: Paragraph 4- From “Enzyme extract of kiwifruit” to “beef sausage ”. Send to “Actinidin”

Response: Para 4 shifted to actinidin.

Comment: Paragraph 5- From “Nam et al explored” to “commercialneutrase”. Send to “Combinations of proteases”

Response: Para 5 shifted to Novel source of plant proteases section.

Comment: Paragraph 6- From “The crude extract of the mango” to “improved beef tenderness”. Send to “Novel source of plant protease”

Response: Para 6 shifted to Novel source of plant proteases section.

Comment: Paragraph 7- From “The soup containing jackfruit (Actocarpus…)” to “potential to be commercialized”. Send to“Novel source of plant protease”

Response: Para 7 shifted to Novel source of plant proteases section

Comment: Paragraph 8- From “By-products of the pineapple” to “meat slices”. Send to “Combinations of proteases”

Response: para 8 shifted to section-combinations of proteases

Comment: Table 3 Send to “Novel source of plant protease”

Response: Table 3 shifted to section Novel source of plant proteases.

Comment: Paragraph 9- From “Thus, plant proteases” to “other quality optimally”. Send to “Plant protease as a natural meat tenderizer”

Response: Para 9 shifted to section- plant proteases as natural meat tenderizers.

Reviewer 3 Report

Title: Application of Plant Proteinase in Meat Tenderization: Recent Trends and Future Prospects

The review article “Application of Plant Proteinase in Meat Tenderization: Recent Trends and Future Prospects” highlighted the importance of plant proteases as efficient meat tenderizers, which reduces the shear force value and degrades the myofibrillar proteins and connective tissue. It is well written review article with some interesting findings; however, there are some corrections:

Line 28: What is meant by “over-tenderization”, authors should describe it in one sentence.

Line 23-25: The sentence “The application of a combination of plant proteases or plant proteases with animal or microbial proteases….” should be removed from the abstract part, it seems an irrelevant information.

Line 21-34: Information provided in the abstract part is not focused. Authors should rewrite the abstract part by focus the title of the study. Different plant proteinase may be described and effects of tenderization on the meat quality may be discussed.

Line 57: Authors should describe the importance of using the tenderization in meat industry i.e., its importance of using in case of low-quality meat cuts?

Line 39-77: I would suggest the authors to revise the introduction part and describe the importance of tenderness in one paragraph (rather than in whole introduction part), second paragraph should be on importance of tenderization, third paragraph should describe the impact of tenderization on meat quality traits and in last paragraph authors should describe the objectives of the current study.

Line 79: How tenderization is different than aging or conditioning, should be described briefly at the start?

Line 84: Authors should describe the mechanism of meat aging or conditioning i.e., describe different intrinsic enzymes involved in this process.

Line 147: Authors should mention the unit of Temperature in Table 1?

Line 147: Table 1: Authors should also add information about Zingibain, Cucumin as meat tenderizer?

Authors should draw a diagram describing the mechanism by which plant proteases act for tenderization? Like their effect on calpain system? I would suggest the authors to read the following manuscript:

·       Meat tenderization mechanism and the impact of plant exogenous proteases: A review (https://doi.org/10.1016/j.arabjc.2020.102967)

Line 670: The conclusion part of the study needs improvement. Furthermore, authors should suggest more guidelines for future research, i.e., which aspect should be focused for future research.

English grammar and sentence structure should be revised and corrected throughout the manuscript.

Author Response

The authors are thankful to the anonymous reviewer for his critical comments and observations. These comments have helped us in improving the quality of the manuscript. We have edited the manuscript accordingly. Further, it is certified that all the issues raised by the reviewer have been incorporated into the revised manuscript. All the changes were marked in RED color text.

A detailed description of reviewer comments and author responses are as follow-

General Comment: The review article “Application of Plant Proteinase in Meat Tenderization: Recent Trends and Future Prospects” highlighted the importance of plant proteases as efficient meat tenderizers, which reduces the shear force value and degrades the myofibrillar proteins and connective tissue. It is well-written review article with some interesting findings; however, there are some corrections:

Response: Authors thank the reviewer for positive and encouraging observations.

Comment: Line 28: What is meant by “over-tenderization”, authors should describe it in one sentence.

Response: Edited as excessive tenderization (mushy texture) due to indiscriminate breakdown of proteins.

Comment: Line 23-25: The sentence “The application of a combination of plant proteases or plant proteases with animal or microbial proteases….” should be removed from the abstract part, it seems irrelevant information.

Response: deleted

Comment: Line 21-34: Information provided in the abstract part is not focused. Authors should rewrite the abstract part by focus the title of the study. Different plant proteinase may be described and effects of tenderization on the meat quality may be discussed

Response: The abstract is revised as per valuable observation of the reviewer.

Comment: Line 57: Authors should describe the importance of using the tenderization in meat industry i.e., its importance of using in case of low-quality meat cuts?

Response: Edited.

Comment: Line 39-77: I would suggest the authors to revise the introduction part and describe the importance of tenderness in one paragraph (rather than in whole introduction part), second paragraph should be on importance of tenderization, third paragraph should describe the impact of tenderization on meat quality traits and in last paragraph authors should describe the objectives of the current study.

Response: The Introduction part is thoroughly revised as per the valuable suggestion by the reviewer in three paragraphs.

Comment: Line 79: How tenderization is different than aging or conditioning, should be described briefly at the start?

Response: Aging has been described briefly at the start of the paragraph.

Comment: Line 84: Authors should describe the mechanism of meat aging or conditioning i.e., describe different intrinsic enzymes involved in this process  

Response: Added as per suggestion.

Comment: Line 147: Authors should mention the unit of Temperature in Table 1?

Response: Added °C

Comment: Line 147: Table 1: Authors should also add information about Zingibain, Cucumin as meat tenderizer?

Response: Information on Zingibain and curcumin has been added. Further in table 2, a new row of cucumin is added.  

Comment: Authors should draw a diagram describing the mechanism by which plant proteases act for tenderization? Like their effect on calpain system? I would suggest the authors to read the following manuscript:  Meat tenderization mechanism and the impact of plant exogenous proteases: A review (https://doi.org/10.1016/j.arabjc.2020.102967)  

Response: Fig 1 has been presented in the manuscript with a note describing the action of plant proteases.

Comment: Line 670: The conclusion part of the study needs improvement. Furthermore, authors should suggest more guidelines for future research, i.e., which aspect should be focused for future research.

Response: The conclusion part has been revised and edited.

Comment: English grammar and sentence structure should be revised and corrected throughout the manuscript.

Response: The manuscript has been duly checked for English grammar and sentence structure. It has been edited and we hope that it will meet the high standard of Foods in its present form.

Round 2

Reviewer 2 Report

The manuscript, after revision by authors, was noticeably enhanced.

Reviewer 3 Report

The manuscript is sufficiently improved to be accepted in the present form for its publication in Foods. I appreciate the efforts of authors.